# Two-dimensional molybdenum carbide 2D-Mo$_2$C as a superior catalyst for CO$_2$ hydrogenation

Hui Zhou [1,3], Zixuan Chen[1], Evgenia Kountoupi[1], Athanasia Tsoukalou [1], Paula M. Abdala [1], Pierre Florian [2], Alexey Fedorov [1✉] & Christoph R. Müller [1✉]

Early transitional metal carbides are promising catalysts for hydrogenation of CO$_2$. Here, a two-dimensional (2D) multilayered 2D-Mo$_2$C material is prepared from Mo$_2$CT$_x$ of the MXene family. Surface termination groups $T_x$ (O, OH, and F) are reductively de-functionalized in Mo$_2$CT$_x$ (500 °C, pure H$_2$) avoiding the formation of a 3D carbide structure. CO$_2$ hydrogenation studies show that the activity and product selectivity (CO, CH$_4$, C$_2$–C$_5$ alkanes, methanol, and dimethyl ether) of Mo$_2$CT$_x$ and 2D-Mo$_2$C are controlled by the surface coverage of $T_x$ groups that are tunable by the H$_2$ pretreatment conditions. 2D-Mo$_2$C contains no $T_x$ groups and outperforms Mo$_2$CT$_x$, β-Mo$_2$C, or the industrial Cu-ZnO-Al$_2$O$_3$ catalyst in CO$_2$ hydrogenation (evaluated by CO weight time yield at 430 °C and 1 bar). We show that the lack of surface termination groups drives the selectivity and activity of Mo-terminated carbidic surfaces in CO$_2$ hydrogenation.

[1] Department of Mechanical and Process Engineering, ETH Zürich, CH 8092 Zürich, Switzerland. [2] CNRS, CEMHTI UPR3079, Université d'Orléans, F-45071 Orléans, France. [3] Present address: Department of Energy and Power Engineering, Tsinghua University, Beijing 100084, China. ✉email: fedoroal@ethz.ch; muelchri@ethz.ch

Earth-abundant early transitional metal carbides, and in particular carbides of Mo and W, feature catalytic properties similar to those of noble metals[1,2]. This property has been exploited for various catalytic processes including Fischer–Tropsch (FT) synthesis[3,4], methane dry reforming[5], water-gas shift (WGS) reaction[6,7], and CO/$CO_2$ hydrogenation[8,9]. Currently, the conversion of captured $CO_2$ into value-added chemicals or fuels is considered a key strategy to mitigate the yet increasing anthropogenic $CO_2$ emissions[10,11], in particular when combined with $H_2$ obtained using renewable energy[12,13]. Depending on the catalyst and the reaction conditions used, thermocatalytic $CO_2$ hydrogenation can give CO, methanol, dimethyl ether (DME), methane, or heavier hydrocarbons[12,14]. In this context, $Mo_2C$ has been reported as a promising catalyst for $CO_2$ hydrogenation, yielding a particularly high selectivity to CO via the reverse water-gas shift (RWGS) reaction[8,9,15–17]. In a recent development, a highly active and selective K-$Mo_2C$/γ-$Al_2O_3$ catalyst for RWGS has been tested on the pilot scale (ca. 1 kg catalyst)[18]. The activity and selectivity of $CO_2$ hydrogenation catalysts based on molybdenum carbide can be affected by the C/Mo ratio in the catalyst[19,20]. According to density functional theory (DFT) calculations, the Mo-terminated surface of β-$Mo_2C$ is more reactive for the dissociation of $CO_2$ than the C-terminated surface[16], and $H_2$ dissociation is most favored on the Mo-terminated (001) facet, compared to the C-terminated or mixed Mo/C-terminated facets[21,22].

MXenes[23–28], i.e., a recently discovered family of two-dimensional (2D) early transition metal carbides, nitrides, or carbonitrides with a formula of $M_{n+1}X_nT_x$ ($n = 1, 2, 3$, X is C and/or N, and $T_x$ are surface oxo, hydroxy, and/or fluoro groups) can be utilized to improve our understanding of the impact of the surface termination groups $T_x$ on the catalytic activity and selectivity of metal-terminated carbide surfaces in $CO_2$ hydrogenation. For instance, delaminated MXene-derived 2D $Mo_2CO_x$ nanosheets dispersed on a silica support were recently shown to feature poor or no activity in the dry reforming of methane if the oxygen coverage is either too low or too high (corresponding to respective Mo oxidation states of ca. +0.2 and +5.5)[29]. Yet the intermediate oxygen coverage of ca. two-thirds of a surface monolayer (Mo oxidation state of ca. +4) provided the highest catalytic activity in DRM, exceeding notably that of bulk β-$Mo_2C$. DFT calculations show that the surface termination groups of MXenes affect the adsorption energies of the reaction intermediates by influencing the density of states of the Fermi level[30]. These results and related literature reports illustrate that controlling the surface density and the type of termination groups is an important factor to consider when advancing and deploying carbide-based catalysts[31,32].

Multilayered crystalline $Mo_2CT_x$ exhibits a high activity for the WGS reaction[33]. The activity of unreduced $Mo_2CT_x$ is higher relative to $Mo_2CT_x$ that has been partially reduced at 500 °C in 10% $H_2$. Interestingly, catalytic tests with ${}^{13}$C-labeled $Mo_2CT_x$ suggested that carbidic carbon exchanges with ${}^{12}$CO (reactant gas) and therefore catalysis also proceeds at the interlayer surface of $Mo_2CT_x$, in addition to the exterior surface. The mentioned reductive defunctionalization of $Mo_2CT_x$ was found to decrease the cell parameter $c$ from ca. 20.51 to ca. 15.63 Å, therefore the lower activity of reduced $Mo_2CT_x$ can be, at least in part, due to increased mass transport limitations into the interlayer space[33].

2D-$Mo_2C$ has attracted significant attention for applications as a superconductor[34,35] or an electrocatalyst[36,37], however, a robust and scalable synthesis protocol for 2D-$Mo_2C$ has not been developed yet. Indeed, typical approaches to yield 2D-$Mo_2C$ exploited so far chemical vapor deposition onto flat substrates and lead to ultra-thin orthorhombic $Mo_2C$[38,39].

Here, enabled by the scalable synthesis of MXenes[40], we report a gram-scale synthesis of a phase-pure multilayered hexagonal 2D-$Mo_2C$ material with only Mo-terminated basal planes. Experimental protocols were developed that allow avoiding the transformation of 2D-$Mo_2C$ into 3D-$Mo_2C$ during the reductive removal of $T_x$ groups on the $Mo_2CT_x$ surface (Fig. 1a). Comparing 2D-$Mo_2C$ and $Mo_2CT_x$ in the catalytic hydrogenation of $CO_2$, we find that the coverage of $Mo_2CT_x$ with surface termination groups affects the activity and selectivity appreciably. For instance, abundant $T_x$ groups on $Mo_2CT_x$ provide surface acidity, reflected in the formation of DME (among other products) due to the dehydration of methanol; DME is not observed on 2D-$Mo_2C$. The latter catalyst features high activity (CO formation rate ca. 6 g h$^{-1}$ g$_{cat}$$^{-1}$) for $CO_2$ hydrogenation and a high selectivity to CO (ca. 94% at 430 °C). 2D-$Mo_2C$ is by a factor of six per mass of catalyst more active for CO formation at 430 °C than the reference β-$Mo_2C$ catalyst and shows no deactivation on stream for more than 100 h.

## Results and discussion

**Synthesis and characterization.** Multilayered $Mo_2CT_x$ was prepared by Ga etching from $Mo_2Ga_2C$ in aqueous HF[33,41]. The layered structure of $Mo_2CT_x$ was confirmed by XRD analysis (parameter $c = 20.636(3)$ Å, Supplementary Fig. 1), consistent with the lack of a Ga signal in the XPS spectra of $Mo_2CT_x$ (Supplementary Fig. 2)[33,42]. Raman spectra indicate that the peak due to $A_{1g}$ ($\omega_6$) Mo−Ga vibrations found at 314 cm$^{-1}$ in $Mo_2Ga_2C$[43] shifts to 253 cm$^{-1}$ in $Mo_2CT_x$, and a new peak emerges at 490 cm$^{-1}$ due to the surface termination groups (Supplementary Fig. 3)[44]. Temperature-programmed oxidation (TPO), performed in a thermogravimetric analyzer, shows that Mo constitutes ca. 73 mass% of $Mo_2CT_x$ (material dried at 80 °C for 24 h), considering that $Mo_2CT_x$ is oxidized to $MoO_3$ at 600 °C under airflow (Supplementary Fig. 4; formation of $MoO_3$ was confirmed by XRD, see Supplementary Fig. 5). The ${}^{95}$Mo VOCS CPMG MAS NMR spectrum of $Mo_2CT_x$ recorded at 20.0 T shows a set of intense and sharp spinning sidebands arising from a combination of chemical shift anisotropy and quadrupolar interaction. Fitting this spectrum with a single component does not provide a satisfactory result for each individual sideband and therefore a second component is needed. A fit that accounts for all spinning sidebands features a major component centered at −1840 ppm accounting for 95.3% of the full intensity, and a second minor component (4.7%) centered at −1895 ppm. Those positions are slightly dependent on temperature but do not change upon lowering the spinning speed and are hence the isotropic chemical shifts of two distinct molybdenum environments. Those two contributions are displayed in green and light blue respectively, with the experimental spectrum in dark blue and the total simulation in red (Fig. 1b). It is likely that the major Mo site centered at −1840 ppm corresponds to terrace Mo sites. The structural assignment of the minor site is currently unclear. Yet we cannot exclude that the minor site is due to the edge Mo atoms of $Mo_2CT_x$ nanosheets.

We have compared the temperature-programmed reduction (TPR) of $Mo_2CT_x$ in 5% $H_2$/Ar and in pure $H_2$ (50 ml min$^{-1}$, ramp 5 °C min$^{-1}$). $Mo_2CT_x$ was heated from 50 to 500 °C, with the off-gas analyzed using a thermal conductivity detector (TCD). We limited the study by 500 °C because of the reported MXene thermal stability limit of ca. 550−600 °C, beyond which the formation of 3D-$Mo_2C$ occurs[33]. Previous results indicated that the reduction of $Mo_2CT_x$ in dilute $H_2$ (up to 20%) defunctionalizes $T_x$ groups only partially[33]. Three $H_2$ consumption peaks, at 179, 280, and 500 °C, are identified using 5% $H_2$/Ar (Supplementary Fig. 6). The low-temperature peak is close to the temperature at which most of the interlayer water desorbs

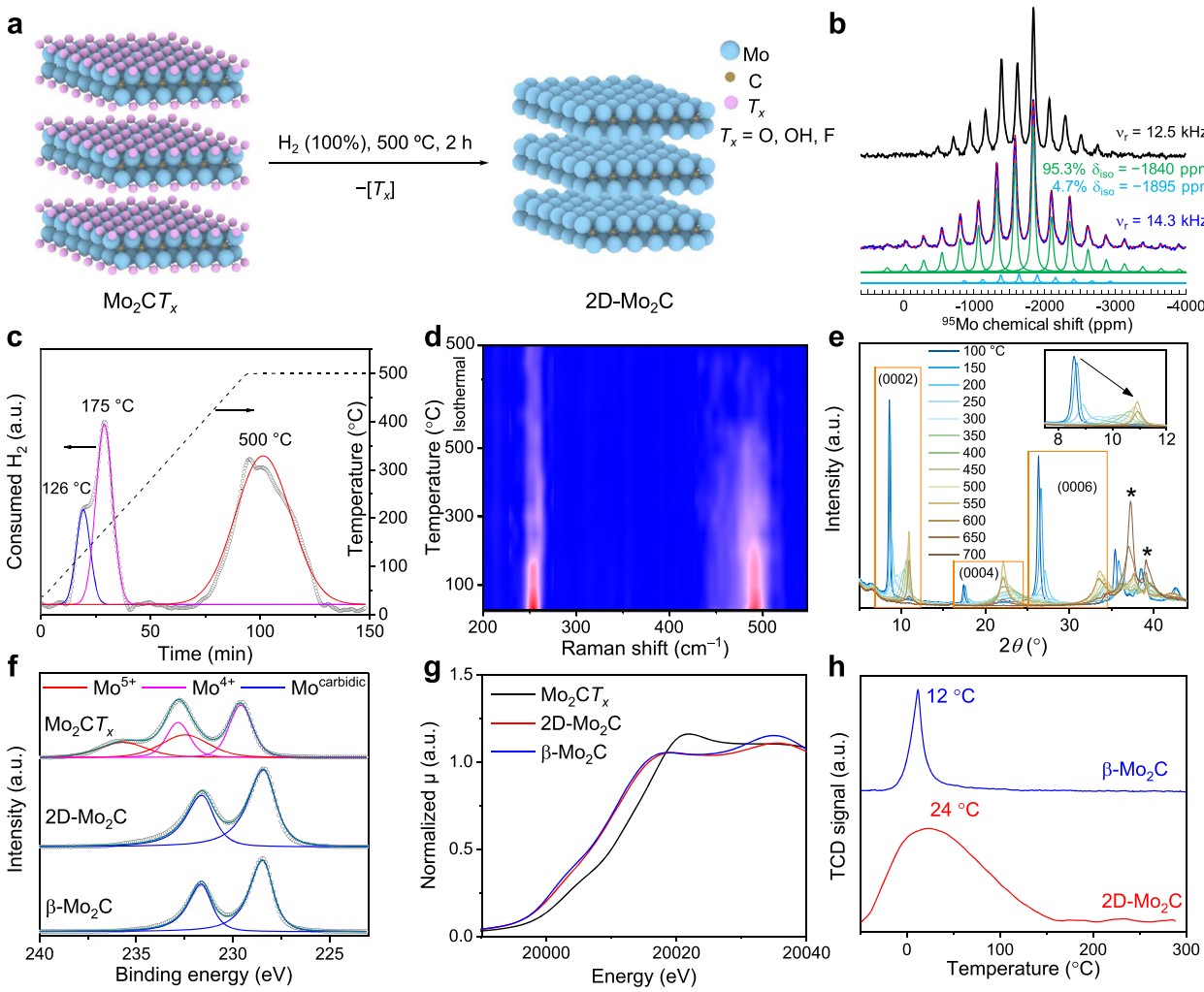

**Fig. 1 Characterization of Mo₂CTₓ and its reduction to 2D-Mo₂C. a** Schematic of the preparation of 2D-Mo₂C from Mo₂CTₓ. **b** $^{95}$Mo VOCS CPMG MAS NMR spectrum of Mo₂CTₓ. **c** Temperature-programmed reduction of Mo₂CTₓ in pure H₂. **d** Reduction of Mo₂CTₓ followed by in situ Raman spectroscopy with a final isothermal heating step at 500 °C performed for 2 h (see Supplementary Fig. 8 for the stacked Raman spectra). **e** Reduction of Mo₂CTₓ followed by in situ XRD (the peaks of β-Mo₂C are marked by asterisks and the main reflections of Mo₂CTₓ are indexed, see Supplementary Fig. 9 for the 3D plot). **f** Mo 3d XPS spectra (see Supplementary Table 1 for the fitting parameters). **g** Mo K-edge XANES spectra of Mo₂CTₓ, 2D-Mo₂C, and reference β-Mo₂C. **h** CO temperature-programmed desorption (TPD) of 2D-Mo₂C and β-Mo₂C.

(ca. 179 °C), according to the data of a thermogravimetric differential scanning calorimetry (TGA-DSC) measurement (Supplementary Fig. 7), and also consistent with TGA-MS results of Mo₂CTₓ in a He atmosphere[45]. The reduction steps at 280 °C likely are associated with the reductive defunctionalization of the surface hydroxyl groups and possibly, fluorine groups, while the peak at 500 °C is related to the removal of surface fluoro and oxo groups[33,45,46]. When using pure H₂, the reduction peaks are observed at lower temperatures of 126 and 175 °C, and the peak at 500 °C is significantly more intense as compared to when using 5% H₂ (Fig. 1c and Supplementary Fig. 6). This result indicates that the extent of defunctionalization of Tₓ groups at 500 °C can be controlled by the H₂ concentration.

An in situ Raman study of the heating of Mo₂CTₓ from room temperature to 500 °C under an H₂ flow (100% H₂, 50 ml min⁻¹, 5 °C min⁻¹), shows that the intensity of the A₁g peak at 253 cm⁻¹ is decreasing with temperature. Interestingly, the E₂g peak at 490 cm⁻¹ (due to Tₓ groups)[44] disappears during the isothermal heat treatment at 500 °C, consistent with the complete removal of the Tₓ groups at this temperature (Fig. 1d and Supplementary Fig. 8). Note that the E₂g peak is preserved during the in situ

Raman study under Ar[44], indicating that H₂ plays a critical role in removing the Tₓ groups.

The in situ reduction of Mo₂CTₓ followed by X-ray powder diffraction (XRD, 5% H₂/N₂, 200 ml min⁻¹, 5 °C min⁻¹) shows that the (0002) peak due to Mo₂CTₓ (8.5° in the as-synthesized material) shifts, at 500 °C, to a higher angle (10.9°, Fig. 1e). This change reflects a decrease in the *c* lattice parameter from 20.64 to 16.21 Å due to the defunctionalization of the Tₓ groups and the removal of intercalated water[33]. The intensity of the (0002) peak decreases starting from ca. 300 °C and increases again at ca. 500 °C, explained by the reestablishing of a long-range order at 500 °C. Compared to Mo₂CTₓ reduced in 5% H₂/N₂, the material reduced in 100% H₂ (500 °C, 2 h, vide infra, exposed to air prior to the XRD measurement) shows a smaller *c* parameter of 15.43 Å (the interlayer distance can be roughly estimated as ca. 5 Å), with the (0002) peak shifting to 11.5°. Moreover, the (0004) to (0002) intensity ratio of Mo₂CTₓ becomes higher when reduced in 100% H₂ at 500 °C (Supplementary Fig. 10), which is related to a change of the coordinates of the Mo atoms[33]. Because of the presence of a crystalline 2D multilayered structure in this reduced material, and the evidence of the complete defunctionalization of the Tₓ groups,

as discussed in detail below, we refer to this material hereafter as 2D-$Mo_2C$. Increasing the reduction temperature further, the intensity of the (0002) peak decreases and vanishes at 700 °C, implying the loss of the two-dimensional multilayered structure. The diffractogram of $Mo_2CT_{x-700}$ matches that of the bulk β-$Mo_2C$ reference (Supplementary Fig. 10).

Next, the reductive defunctionalization of $Mo_2CT_x$ was performed in a flow reactor using undiluted $H_2$ at 500 °C (contact time 0.1 s $g_{cat}$ $mL^{-1}$, a total of 2 h) and the resulted 2D-$Mo_2C$ product was analyzed by X-ray photoelectron spectroscopy (XPS) using an air-tight transfer cell. We observe that the thus obtained 2D-$Mo_2C$ contains only carbidic Mo sites (Mo $3d_{5/2}$ binding energy of 228.4 eV) and its spectrum matches that of the bulk β-$Mo_2C$ reference (also pre-reduced at 500 °C, Fig. 1f, Supplementary Figs. 12, 13, and Supplementary Table 1). Note that it was reported previously that the reduction of $Mo_2CT_x$ using diluted $H_2$ (i.e., 20 vol% $H_2$ in $N_2$) at 500 °C for 1 h defunctionalizes $T_x$ groups only partially, leading to $Mo^{4+}$ and carbidic Mo states (Mo $3d_{5/2}$ binding energies at 229.3 and 228.5 eV, respectively)[33]. While a fluorine signal can be clearly observed in the F 1s XPS data of $Mo_2CT_x$, $H_2$ treatment in 100% $H_2$ at 500 °C for 2 h decreases the F signal in 2D-$Mo_2C$ to the noise level, consistent with the deep removal of the F groups (Supplementary Figs. 13 and 14). The fitted C 1s region in 2D-$Mo_2C$ reveals no C—O and COO features and contains only Mo–C and C–C features (vide infra). The control experiment shows that $Mo_2CT_x$ reduced using 10% $H_2/N_2$ and under otherwise identical conditions does not give 2D-$Mo_2C$ owing to the incomplete defunctionalization of $T_x$ groups, according to XPS data (Supplementary Fig. 15 and Supplementary Table 1)[33].

Scanning electron microscopy (SEM) reveals a similar multilayered hexagonal microstructure for $Mo_2CT_x$ and 2D-$Mo_2C$ (average hexagonal radius and thickness are 1.0 and 0.3 μm, respectively); in contrast, β-$Mo_2C$ shows no hexagonal nanoplatelets (Supplementary Figs. 16−19). An intense fluorine signal in $Mo_2CT_x$ and the lack of thereof in 2D-$Mo_2C$ are revealed by scanning transmission electron microscopy (STEM) energy-dispersive X-ray analysis (STEM-EDX) (Supplementary Fig. 20). A selected area electron diffraction (SAED) pattern of 2D-$Mo_2C$ displays the Mo-terminated (0002) plane as the main exposed facet, i.e., 2D-$Mo_2C$ features an almost completely Mo-terminated surface (Supplementary Fig. 21). A similar SAED result is observed for $Mo_2CT_{x-700}$, indicating that sintering of the individual layers and formation of the β-$Mo_2C$ structure (according to XRD results discussed above) retains the Mo-terminated surface (Supplementary Fig. 22). In contrast, SAED of β-$Mo_2C$ shows (0002) and (11$\bar{2}$0) facets (Supplementary Fig. 23), indicating that both Mo-terminated and C-terminated planes are exposed, consistent with a previous study of β-$Mo_2C$[16].

The Mo K-edge X-ray absorption near edge structure (XANES) spectrum of 2D-$Mo_2C$ is clearly different from that of $Mo_2CT_x$ and similar to that of β-$Mo_2C$ (edge positions 20001.4, 20011.1, and 20000.8 eV, corresponding to the oxidation state of Mo of +0.5, +3.9, and +0.3, respectively, Fig. 1g and Supplementary Fig. 24)[33,47]. While oxidation states of Mo in 2D-$Mo_2C$ and β-$Mo_2C$ are very close, there are changes in the post white line region, likely owing to the different structures of these two carbides (Fig. 1g).

The extended X-ray absorption fine structure (EXAFS) data of β-$Mo_2C$ can be fitted with a coordination number (CN) to the

nearest carbon in the Mo–C shell of 3 and a CN of 6 in the nearest Mo–Mo shell (Supplementary Table 2 and Supplementary Figs. 25, 26). The Mo–Mo shell in 2D-$Mo_2C$ has a CN of 6.1(7) and a distance of 2.95(1) Å, which are comparable with that of β-$Mo_2C$. A CN in Mo–C shell in 2D-$Mo_2C$ is 2.6(5), i.e., comparable to β-$Mo_2C$ but significantly lower than in $Mo_2CT_x$ (7(1) for Mo–C/$T_x$ shell). Overall, XANES and EXAFS results are consistent with the formation of 2D-$Mo_2C$ from $Mo_2CT_x$. To summarize, all characterization data discussed above suggest 2D-$Mo_2C$ can be obtained from $Mo_2CT_x$ selectively, i.e., avoiding the formation of 3D-$Mo_2C$, if optimized conditions for the complete reductive defunctionalization of $T_x$ groups in $Mo_2CT_x$ are used.

We performed CO temperature-programmed desorption (TPD) experiments to compare the properties of surface sites in 2D-$Mo_2C$, $Mo_2CT_x$, and the β-$Mo_2C$ reference. $Mo_2CT_x$ does not absorb CO (Supplementary Fig. 27). β-$Mo_2C$ features a sharp CO desorption peak at 12 °C, indicating uniform Mo sites (Fig. 1h). In contrast, 2D-$Mo_2C$ shows a broad CO desorption peak at 24 °C, possibly due to mass transfer effects arising from CO molecules adsorbed also inside the 2D pores of this multilayered material. $H_2$ TPD results of β-$Mo_2C$ and 2D-$Mo_2C$ were also compared. While both materials display well-defined low-temperature desorption peaks centered at −19 and −14 °C for, respectively, β-$Mo_2C$ and 2D-$Mo_2C$, the latter material also has broad $H_2$ desorption peaks at higher temperatures (ca. 140 and 352 °C) explained by the two-dimensional structure of this carbide and different nature of surface sites in 2D-$Mo_2C$ (Supplementary Fig. 28).

Subsequently, CO chemisorption experiments were performed to compare the quantity of exposed Mo sites in the materials under investigation. The CO chemisorption capacity of $Mo_2CT_x$ is very low (0.2 μmol $g^{-1}$), due to abundant surface termination groups (Table 1). After $H_2$ treatment at 300 °C, the amount of chemisorbed CO increases to 14.4 μmol $g^{-1}$, indicating the partial removal of the $T_x$ groups. 2D-$Mo_2C$ (prepared in situ prior to CO chemisorption analysis) shows a significantly increased CO capacity of 41.1 μmol $g^{-1}$, which exceeds the CO capacity of β-$Mo_2C$ (also reduced in $H_2$ at 500 °C in situ) by ca. a factor of eight (Table 1). This result is explained by the larger specific surface area of the exposed Mo-terminated facets in 2D-$Mo_2C$, which is also consistent with the similar $CO_2$ adsorption energy of MXenes and 3D transitional metal carbides[48]. $H_2$ treatment of $Mo_2CT_x$ at 700 °C results in a material with a lower CO capacity (12.8 μmol $g^{-1}$) compared with that of 2D-$Mo_2C$ due to the sintering of the layered structure of 2D-$Mo_2C$, as indicated by XRD, yet the CO capacity is still higher than that of the reference β-$Mo_2C$ (5.0 μmol $g^{-1}$).

**Catalytic performance.** The catalytic performance of the prepared materials for $CO_2$ hydrogenation was evaluated first at 230 °C and 25 bar ($H_2/CO_2/N_2$ = 3/1/1). The main products obtained using $Mo_2CT_x$ are CO, $CH_4$, and methanol (intrinsic formation rates of 53, 16, and 13 mg $h^{-1}$ $g_{cat}^{-1}$ and intrinsic selectivities of 54, 23, and 11%, respectively) with DME and $C_2$–$C_5$ hydrocarbons as minor products (Fig. 2a and Supplementary Fig. 29). The product formation rates are much higher than those from previous studies on the $CO_2$ reduction by MXene-based electro[49] or photocatalysis[50]. All intrinsic formation rates and intrinsic selectivities reported in

**Table 1 CO uptake capacities determined by pulse chemisorption measurements.**

| Material | $Mo_2CT_x$ | $Mo_2CT_{x-TOS1h}$ | $Mo_2CT_{x-300}$ | 2D-$Mo_2C$ | $Mo_2CT_{x-700}$ | β-$Mo_2C$ |
|---|---|---|---|---|---|---|
| CO uptake (μmol $g^{-1}$) | 0.2 | 4.6 | 14.4 | 41.1 | 12.8 | 5.0 |

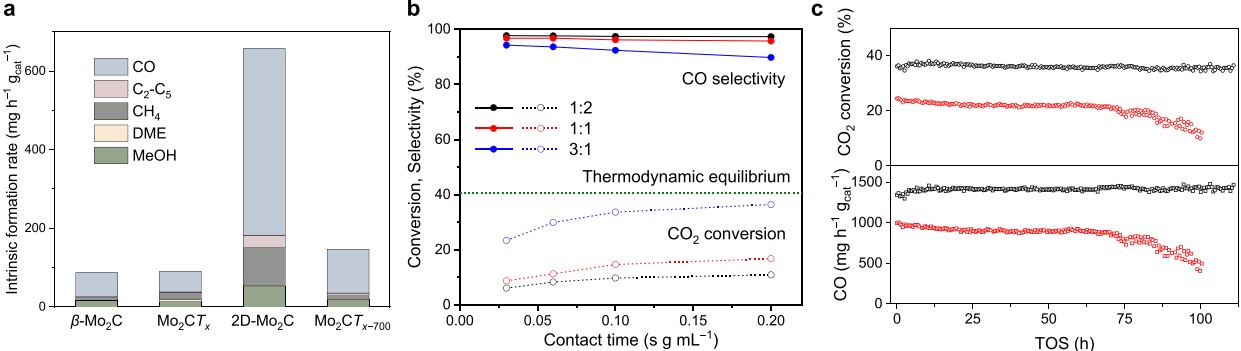

**Fig. 2 Results of $CO_2$ hydrogenation tests. a** Intrinsic formation rates obtained after $H_2$ pretreatment of $Mo_2CT_x$ at different temperatures. Reaction conditions: 230 °C, 25 bar, $H_2/CO_2/N_2 = 3/1/1$. Intrinsic formation rates were obtained by extrapolation to zero conversion (zero contact time, see Supplementary Fig. 29) and $CO_2$ conversion was in the range 0.3−8.5%. **b** Hydrogenation of $CO_2$ at 430 °C and 1 bar with 2D-$Mo_2C$ under variable $H_2:CO_2$ ratios. The green dash line shows the calculated thermodynamic equilibrium (based on the minimization of the Gibbs free energy at 430 °C, 1 bar, $H_2/CO_2/N_2 = 3/1/1$). **c** Stability test of 2D-$Mo_2C$ (black) and industrial Cu-ZnO-$Al_2O_3$ (red) at ca. 100 h of TOS (430 °C, 1 bar, $H_2/CO_2/N_2 = 3/1/1$, contact time 0.2 s $g_{cat}$ $mL^{-1}$).

this work (Supplementary Table 3) are derived from contact time studies using extrapolation to zero conversion (i.e., zero contact time). The low CO chemisorption capacity of $Mo_2CT_x$ increases significantly after a 1-h exposure to an $H_2/CO_2$ mixture (3/1, 1 bar), reaching 4.6 µmol $g^{-1}$ (Table 1). This result indicates that the $T_x$ groups in $Mo_2CT_x$ are partially removed under $CO_2$ hydrogenation conditions. A minor amount of DME (8% selectivity) is formed on $Mo_2CT_x$; this product is not observed when using the β-$Mo_2C$ or 2D-$Mo_2C$ catalysts (Fig. 2a and Supplementary Fig. 30). DME is a typical product of methanol dehydration on Brønsted or strong Lewis acid sites[51–53], which suggests strong acidity in $Mo_2CT_x$. Indeed, the $NH_3$ TPD experiment reveals intense $NH_3$ desorption peaks at ca. 120 and 500 °C in $Mo_2CT_x$ that had been exposed to an $H_2/CO_2$ mixture (3/1, 1 bar) for 1 h (Supplementary Fig. 31). Interestingly, while the related MXene material, $Ti_3C_2T_x$, has been reported to give no $NH_3$ desorption peaks, Brønsted and Lewis acidity was found in $Ti_3C_2T_x$ using pyridine as a probe molecule[54]. Brønsted acidity of highly oxophilic $Mo_2C$ has been reported previously to appear after the exposure of $Mo_2C$ to oxygen; in contrast, no significant acidity is observed for reduced β-$Mo_2C$[55], which is also consistent with the $NH_3$ TPD results of materials of this study, i.e., β-$Mo_2C$, 2D-$Mo_2C$, and $Mo_2CT_{x-TOS1h}$ (Supplementary Fig. 31).

$H_2$ pretreatment at 300 °C (100% $H_2$, 2 h) does not change appreciably the activity and selectivity of $Mo_2CT_x$ (Supplementary Fig. 29). In contrast, $H_2$ pretreatment at 500 °C (2 h) yields 2D-$Mo_2C$ and impacts the catalytic performance significantly (Fig. 2a). In particular, the intrinsic methanol formation rate for 2D-$Mo_2C$ is 53 mg $h^{-1}$ $g_{cat}$$^{-1}$, which is comparable to the activity of the Cu-ZnO-$ZrO_2$ catalyst (37 wt% Cu loading, ca. 45 mg $h^{-1}$ $g_{cat}$$^{-1}$) in similar (220 °C, 30 bar, $H_2/CO_2 = 3/1$) conditions[56]. The formation rates of CO and $C_2$–$C_5$ hydrocarbons increase as well for 2D-$Mo_2C$, reaching 475 and 32 mg $h^{-1}$ $g_{cat}$$^{-1}$; these values are ca. 9 and 19 times higher than those obtained using $Mo_2CT_x$ (Supplementary Table 3). Thus, the higher activity of 2D-$Mo_2C$ relative to $Mo_2CT_x$ correlates with the trend in CO chemisorption as discussed above (Table 1). Noteworthy, the activity observed for 2D-$Mo_2C$ is higher than for $Mo_2CT_x$ despite the reduced interlayer distance in 2D-$Mo_2C$ (unit cell parameters $c$ is 15.43 and 20.64 Å in 2D-$Mo_2C$ and $Mo_2CT_x$, respectively; 2D-$Mo_2C$ was exposed to air prior to the XRD measurement). This result is also different from that found in a recent DRM study, where an oxycarbidic 2D-$Mo_2CO_x$ with an average Mo oxidation state of +4 (modeled by the 2/3 of a monolayer oxygen surface coverage) is the active phase[29].

Interestingly, our previous study showed that $Mo_2CT_x$ is more active in the WGS reaction (CO + $H_2O$ ↔ $CO_2$ + $H_2$) than $Mo_2CT_x$ pretreated under 10% $H_2/N_2$ at 500 °C for 1 h (i.e., only a partially reduced material)[33]. We have suggested that the reduced interlayer distance of the pretreated catalyst may lead to increased mass transport limitations in the interlayer space[33]. In this study, we have compared the activity of $Mo_2CT_x$ and 2D-$Mo_2C$ for WGS at 500 °C (CO/$H_2O$/$N_2$ = 1/1/9). $Mo_2CT_x$ shows a stable activity for at least 10 h of TOS, consuming CO at a rate of ca. 5.6 g $h^{-1}$ $g_{cat}$$^{-1}$ (>99% selectivity to $CO_2$ at 49% conversion of CO). However, while the initial activity of 2D-$Mo_2C$ is similar to that of $Mo_2CT_x$ (ca. 4.8 g $h^{-1}$ $g_{cat}$$^{-1}$), in the first 4 h of TOS it decreases to ca. 2.8 g $h^{-1}$ $g_{cat}$$^{-1}$ of consumed CO (>99% selectivity to $CO_2$ at 25% conversion of CO, Supplementary Fig. 32). XANES data reveals that 2D-$Mo_2C$ is oxidized under the WGS conditions used, reaching after 10-h of TOS an oxidation state (and therefore a $T_x$ surface coverage) close to that of $Mo_2CT_x$ (Supplementary Fig. 33). Owing to this similar $T_x$ surface coverage, the lower activity of 2D-$Mo_2C$ after 4-h of TOS is consistent with mass transport limitations due to the reduced interlayer distance in the $H_2$-pretreated $Mo_2CT_x$ materials. The decrease of the activity of 2D-$Mo_2C$ in the first 4 h of TOS is therefore linked with the slow surface oxidation (likely, by steam) of 2D-$Mo_2C$, including the interlayer Mo sites. Note that such significant oxidation of 2D-$Mo_2C$ under WGS conditions contrasts substantially the only minor oxidation of 2D-$Mo_2C$ in RWGS conditions (vide infra).

2D-$Mo_2C$ shows a ca. eight times higher intrinsic $CO_2$ hydrogenation reaction rate normalized per catalyst mass compared to β-$Mo_2C$ (Supplementary Table 3). In other words, the intrinsic rates of $CO_2$ consumption normalized by surface Mo sites determined by CO chemisorption in 2D-$Mo_2C$ and β-$Mo_2C$ are similar. This is in sharp contrast with the significantly higher specific activity of 2D-$Mo_2CO_x$ relative to β-$Mo_2C$ in the dry reforming of methane[29]. If we compare the intrinsic product formation rates normalized by surface Mo sites, the rate of CO formation is also similar on 2D-$Mo_2C$ and β-$Mo_2C$. These results indicate that the increased activity of 2D-$Mo_2C$ (per mass catalyst) is mostly due to the greater fraction of exposed Mo atoms available for the catalytic reaction in the 2D material. However, the formation rate normalized by surface Mo of $C_1$–$C_5$ is higher on 2D-$Mo_2C$ relative to β-$Mo_2C$ (33 vs. 18 g $h^{-1}$ $g_{surf(Mo)}$$^{-1}$), which is offset by the lower methanol formation rate (14 vs. 35 g $h^{-1}$ $g_{surf(Mo)}$$^{-1}$, Supplementary Fig. 34). This difference suggests that other factors are also in play, possibly

related to differences in the adsorption energies of reactive intermediates in these two catalysts.

We have performed a control experiment and compared the activities of 2D-Mo$_2$C and the material obtained after the pretreatment of Mo$_2$CT$_x$ in 10% H$_2$ at 500 °C for 2 h. The latter catalyst denoted Mo$_2$CT$_{x\text{-}500\text{-}10\%}$ shows lower formation rates of hydrogenation products (Supplementary Fig. 35). This inferior activity is explained by the incomplete defunctionalization of $T_x$ groups in Mo$_2$CT$_{x\text{-}500\text{-}10\%}$ as confirmed by XPS data (Supplementary Fig. 15). Furthermore, a reduction of Mo$_2$CT$_x$ under pure H$_2$ at 700 °C leads to a notably lower catalytic activity (Fig. 2a), as compared to 2D-Mo$_2$C, explained by the formation of 3D-Mo$_2$C (Supplementary Fig. 10). This trend is in agreement with the CO chemisorption results (Table 1).

The catalytic performance of Mo$_2$CT$_x$ and 2D-Mo$_2$C was then compared at three additional temperatures, i.e., 130, 330, and 430 °C at 25 bar using an H$_2$:CO$_2$ ratio of 3. The results show that the distribution of products on Mo$_2$CT$_x$ and 2D-Mo$_2$C depends on the reaction temperature, in agreement with the thermodynamic calculations (Supplementary Figs. 36−39). More specifically, at 130 °C, a high methanol selectivity of 41 and 62% is observed for Mo$_2$CT$_x$ and 2D-Mo$_2$C, respectively. Increasing the reaction temperature from 130 to 430 °C, increases the intrinsic CO selectivity for Mo$_2$CT$_x$ from 44 to 91%, yet the CO selectivity for 2D-Mo$_2$C increases only from 31% at 130 °C to 65% at 230 °C, before it decreases again to 35 and 39% at 330 °C and 430 °C, respectively. (Supplementary Fig. 39). In contrast, the intrinsic selectivity to C$_1$−C$_5$ hydrocarbons increases from 6% at 130 °C to 61% at 430 °C with 2D-Mo$_2$C. Among the C$_1$−C$_5$ hydrocarbons, CH$_4$ is the major component with a partial selectivity of 83% among the C$_1$−C$_5$ products; C$_2$−C$_5$ hydrocarbons are predominantly alkanes (>99%, Supplementary Fig. 40). Given the known activity of Mo$_2$C-based catalysts in the FT process[3], the formation of C$_2$−C$_5$ hydrocarbons can be explained by the FT activity of 2D-Mo$_2$C with H$_2$ and CO formed in situ by the hydrogenation of CO$_2$. We note that the hydrogenation of propene (formed in situ from propyne) to propane has been reported to occur on molybdenum carbides above 300 °C[57].

Next, contact time studies were also performed at lower pressures, i.e., 1 bar and 5 bar, using 230 °C and an H$_2$:CO$_2$ ratio of 3. At 1 bar, CO is the main product for both Mo$_2$CT$_x$ and 2D-Mo$_2$C (73 and 90% selectivity, respectively), in agreement with thermodynamic calculations (Supplementary Figs. 36, 41−43). Increasing the reaction pressure leads to an increase in the formation rates of all products (Supplementary Figs. 41 and 42), while the selectivity to CO decreases. The highest selectivity to CO is obtained at a low H$_2$:CO$_2$ ratio of 1:2, and increasing the H$_2$:CO$_2$ ratio increases the selectivities to other products (methanol, DME, and hydrocarbons, see Supplementary Figs. 44−46). Overall, at all conditions tested, the intrinsic formation rates for the different products are higher on 2D-Mo$_2$C relative to Mo$_2$CT$_x$ except for DME, which is not observed on 2D-Mo$_2$C.

Based on the above results and those of the thermodynamic calculation (Supplementary Fig. 36), we performed a CO$_2$ hydrogenation test at 430 °C and 1 bar to evaluate the maximized weight time yield (WTY) of CO. In these conditions, the selectivity to CO is high (90−99%) for both Mo$_2$CT$_x$ and 2D-Mo$_2$C, with the selectivities depending on the H$_2$:CO$_2$ ratio tested (1:2, 1:1, 3:1) and the contact time (Fig. 2b and Supplementary Fig. 47). A high CO formation rate of ca. 6 g h$^{-1}$ g$_{cat}^{-1}$ is obtained with 2D-Mo$_2$C at 0.03 s g$_{cat}$ mL$^{-1}$ contact time and an H$_2$:CO$_2$ ratio of 3:1, i.e., six times higher than obtained with β-Mo$_2$C (Supplementary Fig. 48). Keeping these conditions the same, but changing the contact time to 0.2 s g$_{cat}$ mL$^{-1}$, yields a CO$_2$ conversion that is close to the thermodynamic equilibrium

(Fig. 2b). This WTY of CO exceeds reported values for the RWGS reaction on Cu-Mo$_2$C (Cu loading 1 wt%) at similar reaction conditions[58] or the benchmark Cu-ZnO-Al$_2$O$_3$ (Cu loading ca. 60 wt%) catalyst (Supplementary Fig. 48) tested at identical reaction conditions, indicating a remarkable catalytic activity of 2D-Mo$_2$C for the RWGS. We also performed a catalytic test of 2D-Mo$_2$C at 430 °C and 25 bar. In contrast to β-Mo$_2$C or the Cu-ZnO-Al$_2$O$_3$ catalysts, 2D-Mo$_2$C can be made selective for methane, i.e., >80% selectivity to methane can be achieved at a long contact time of 2.4 s g$_{cat}$ mL$^{-1}$ (Supplementary Fig. 49). Therefore, 2D-Mo$_2$C is a more versatile catalyst for CO$_2$ hydrogenation compared to the conventional β-Mo$_2$C or Cu-ZnO-Al$_2$O$_3$ catalysts.

Turning to catalytic stability, the catalytic performance of Mo$_2$CT$_x$ is stable over 36 h of TOS at 230 °C (25 bar, H$_2$/CO$_2$/N$_2$ = 3/1/1, Supplementary Fig. 50). However, after 36 h of TOS at a higher temperature of 430 °C (material denoted Mo$_2$CT$_{x\text{-}TOS36h(430)}$), the catalytic activity of Mo$_2$CT$_x$ changes significantly when the reaction temperature is decreased to 230 °C. The formation rate of CO increases from 62 to 116 mg h$^{-1}$ g$_{cat}^{-1}$, and the formation rate of C$_2$−C$_5$ increases from 4 to 10 mg h$^{-1}$ g$_{cat}^{-1}$. DME is not observed at 230 °C after 36 h of TOS at 430 °C, indicating the loss of acidity of the catalyst. XRD result shows that the $c$ parameter decreases to 15.41 Å (Supplementary Fig. 51), indicating that the multilayered material undergoes a reduction in the interlayer spacing. Mo K-edge XANES shows that the Mo white line and the edge position are shifted to lower energies after reaction at 430 °C (Supplementary Fig. 52), in agreement with the reduction of Mo. Likewise, Mo 3$d$ XPS also shows that the material is reduced in situ during the reaction at 430 °C, with the Mo$^{5+}$ fraction decreasing from 46 to 31% and the carbidic Mo fraction increasing from 0 to 8% (Supplementary Fig. 53 and Supplementary Table 1). This indicates that the activity change of Mo$_2$CT$_{x\text{-}TOS36h(430)}$ is caused by the reduction of Mo with time on stream. Subjecting Mo$_2$CT$_{x\text{-}TOS36h(430)}$ to 100% H$_2$ at 500 °C for 2 h further increases its activity, making it very close to that of 2D-Mo$_2$C (Supplementary Fig. 54).

2D-Mo$_2$C shows no deactivation after more than 100 h TOS at 430 °C (1 bar, H$_2$/CO$_2$/N$_2$ = 3/1/1), with a stable CO$_2$ conversion and CO formation rate. In contrast, the industrial Cu-ZnO-Al$_2$O$_3$ catalyst deactivates in these conditions by ca. 50% (Fig. 2c), likely due to the oxidation of Cu and/or agglomeration of ZnO. XRD analysis of the used catalyst shows no change of the multilayered structure of the 2D-Mo$_2$C (Supplementary Fig. 55). Operando Raman study shows that there is no reappearance of a band at 490 cm$^{-1}$ due to surface termination groups, as was observed in the as-synthesized Mo$_2$CT$_x$ discussed above (Supplementary Fig. 56). However, Mo K-edge XANES data show that the edge position increases from 20001.4 to 20002.0 eV after reaction (Supplementary Fig. 57), indicating that Mo becomes slightly more oxidized, i.e., the average oxidation state of Mo in used 2D-Mo$_2$C is slightly higher relative to that in fresh 2D-Mo$_2$C (Supplementary Fig. 24). This is also confirmed by $^{13}$C MAS NMR analysis of the used 2D-Mo$_2$C (vide infra).

**Mechanistic study.** Decoupling of the oxidation and reduction steps in CO$_2$ hydrogenation (i.e., operation in a chemical looping scheme) can be beneficial for the separation of products and an improved energy integration due to operation in separate exo/endothermic half reactions[59]. To investigate whether the oxidation and reduction steps can be decoupled for the net CO$_2$ hydrogenation reaction, we flowed 2% CO$_2$ in N$_2$ through a fixed bed of 2D-Mo$_2$C (100 mg) at 430 °C and 1 bar and detected that 294 µmol g$^{-1}$ of CO has been formed in 10 min (Fig. 3a). After regeneration with 10% H$_2$ at 430 °C for 10 min, a lower CO amount

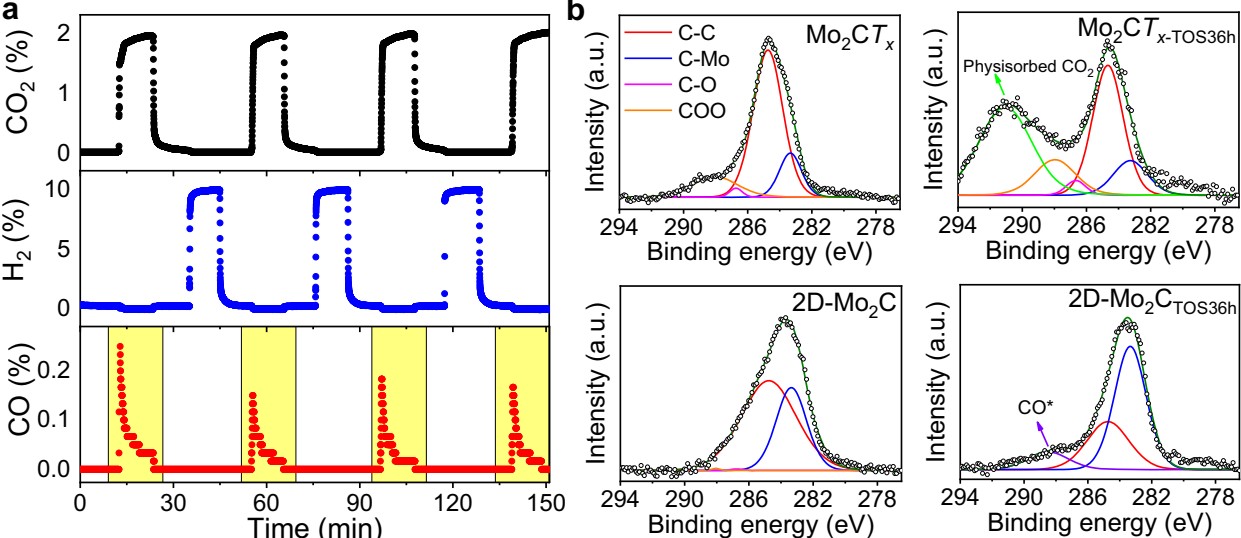

**Fig. 3 Sequential CO₂ hydrogenation study. a** CO₂ dissociation study on 2D-Mo₂C (430 °C, 1 bar). **b** C 1s XPS regions of fresh and used Mo₂CTₓ and 2D-Mo₂C catalysts, i.e., before and after their exposure to the reaction stream.

is detected in the second cycle (176 µmol g⁻¹, Supplementary Table 4), indicating that 2D-Mo₂C cannot be fully recovered in 10% H₂. No further deactivation is observed from the second to the fourth cycle. A similar catalytic test was then performed in a TGA, where the reduction and oxidation of the catalyst could be monitored by the mass change (Supplementary Fig. 58). The sample mass after the second H₂ treatment is 0.5 wt% higher compared to that after the original H₂ reduction, providing further evidence for the partial oxidation of Mo (formation of 2D-Mo₂CO$_x$) with TOS, as also suggested by the Mo K-edge XAS data. In sharp contrast, the amounts of CO formed under identical chemical looping conditions but using β-Mo₂C was below the detection limit of the infrared detector (Supplementary Fig. 59). While these chemical looping type experiments demonstrate that H₂ is not required to be involved in the CO formation step from CO₂ on 2D-Mo₂C, we cannot exclude that CO formation under RWGS conditions involves formate intermediates[60].

¹³C MAS NMR data of the used (and still active) 2D-Mo₂C catalyst after 9 h of TOS (430 °C, 1 bar, H₂/CO₂/N₂ = 3/1/1) was recorded (Supplementary Fig. 60). The decreased intensity of the carbon peak at 175 ppm (oxycarbide carbon) and the appearance of a new broad peak at 275 ppm (carbidic carbon) confirms that 2D-Mo₂C is oxidized slightly under the reaction conditions[33,61]. However, the broad peak at 275 ppm may also contain a contribution from surface C* species. The ⁹⁵Mo NMR spectrum of the used 2D-Mo₂C catalyst was difficult to obtain since the compound did not spin in a 20.0 T magnetic field; this behavior is characteristic for materials with metallic conductivity. Recording the spectrum under static conditions yields a very broad spectrum spanning almost 5000 ppm (Supplementary Fig. 61). This observation is fully consistent with the very broad line observed for a Mo/ZSM-5 catalyst at an ultra-high field and is assigned to Mo₂C/MoO$_x$C$_y$ species[62]. A fit of the spectrum (neglecting Knight shifts) estimates the isotropic chemical shift at 615 ± 100 ppm, in good agreement with the 900 ppm value found for Mo₂C[63]. We observe a quadrupolar coupling constant C$_Q$ of ca. 20.6 ± 0.1 MHz, which is much larger than the C$_Q$ of 6.1 MHz found for Mo₂C. In addition, we find in our used 2D-Mo₂C catalyst that a quadrupolar asymmetry parameter η$_Q$ = 0.0 ± 0.15, as opposed to η$_Q$ = 0.98 that is found in Mo₂C. This low η$_Q$ value is consistent with a cylindrical symmetry of the Mo environment in low-dimensional Mo sites (i.e., 2D-morphology) in used 2D-Mo₂C.

To assess the nature of carbonaceous surface species in working catalysts, we compared the C 1s XPS data of Mo₂CTₓ and 2D-Mo₂C catalysts before and after their exposure to the reaction stream at 430 °C, 1 bar, 36 h (Fig. 3b and Supplementary Figs. 62, 63). CO₂ physisorbed on Mo₂CTₓ is identified by a peak with a binding energy of ca. 291.0 eV[8]. As such peak is not observed on 2D-Mo₂C it is suggested that CO₂ is converted rapidly on the surface of 2D-Mo₂C under reaction conditions; instead, surface CO species (low intensity feature with a fitted maximum position at 288.9 eV) are observed on 2D-Mo₂C after exposure to the reaction stream.

To conclude, we have shown that the activity and selectivity of multilayered Mo₂CTₓ catalysts under CO₂ hydrogenation conditions can be tuned by controlling the surface termination groups through H₂ treatment. A Tₓ-free, Mo-terminated 2D-Mo₂C is synthesized by treatment in pure H₂ at 500 °C. 2D-Mo₂C is more active for the RWGS reaction compared to β-Mo₂C and the Cu-ZnO-Al₂O₃ reference. With the CO yield close to the thermodynamic equilibrium at 430 °C (that is in harsh, H₂O-reach conditions), the 2D-Mo₂C is stable for more than 100 h TOS under the studied reaction conditions. As the exposed terrace surface of 2D-Mo₂C is purely Mo-terminated, its CO chemisorption capacity exceeds that of β-Mo₂C by a factor of ca. eight, leading in turn to a comparable increase of its activity in the hydrogenation of CO₂ to CO. 2D-Mo₂C is highly active for CO₂ dissociation and can also be used in the chemical looping CO₂-H₂ redox cycles. Overall, the results obtained indicate that 2D-Mo₂C is a promising catalyst for CO₂ hydrogenation, exhibiting a remarkable catalytic activity and stability.

## Methods

**Material synthesis.** Mo₂Ga₂C was synthesized following a previously reported method[33,64,65]. β-Mo₂C (1 g, Sigma-Aldrich, 99.5 wt%) was mixed with metallic gallium (3 g, Alfa Aesar, 99.99 wt%) at 45 °C to obtain a mirror-like paste. The paste was flame-sealed under dynamic vacuum (ca. 10⁻⁵ mbar) and annealed at 850 °C for 48 h. The solid was then stirred in 12 M HCl (20 mL, VWR Chemicals) at room temperature for 48 h and washed with water until a pH of ca. 6 was reached. The powder was then dried at 80 °C overnight to obtain Mo₂Ga₂C.

Mo₂CTₓ was prepared by stirring Mo₂Ga₂C (1 g) in 50 mL of HF solution (14 M, Sigma-Aldrich) at 400 rpm in a sealed Teflon-lined autoclave for 10 days at 140 °C[33]. It should be noted that experiments with highly toxic concentrated HF solutions should be conducted in a dedicated fume hood and require extra care. After washing with deionized water until a pH of ca. 6 is reached and drying at 80 °C overnight the Mo₂CTₓ powder was obtained. Mo₂CTₓ₋₃₀₀ and 2D-Mo₂C

were prepared by treating the $Mo_2CT_x$ at 300 and 500 °C for 2 h (heating rate 5 °C min$^{-1}$) under 100% $H_2$, respectively.

The industrial CuO-ZnO-Al$_2$O$_3$ catalyst (63.5 wt% CuO, 25 wt% ZnO, 10 wt% Al$_2$O$_3$, and 1.5 wt% MgO fume) was obtained from Alfa Aesar and reduced at 500 °C before the catalytic test. All the materials were kept in a glovebox prior to characterization and catalytic tests. Except for XRD, ex situ characterization data of the activated materials have been acquired in pristine conditions, i.e., without exposure to air.

**Material characterization.** Powder X-ray diffraction (XRD) data were collected on a PANalytical Empyrean X-ray diffractometer with a Bragg-Brentano HD mirror operated at 45 kV and 40 mA using Cu K$\alpha$ radiation ($\lambda = 1.5418$ Å). The materials were scanned in the $2\theta$ range of 5–90° using the step size of 0.0167° and a scan time per step of 3 s. In situ XRD was performed in the same instrument using an Anton Paar XRK 900 reactor chamber, in the range of 5–45° from room temperature to 700 °C (5 °C min$^{-1}$) under 5% H$_2$/Ar. XPS was performed on a Sigma 2 instrument (Thermo Fisher Scientific) with a UHV chamber (nonmonochromatic 200 W Al K$\alpha$ source, a hemispherical analyzer, and a seven-channel electron multiplier). The analyzer-to-source angle was 50° and the emission angle was 0°. An air-tight cell was used to transfer samples (supported on carbon tapes) from the glovebox to the XPS chamber without exposure to air[66].

The X-ray absorption spectra at the Mo K-edge were measured at the SuperXAS beamline at the Swiss Light Source (Paul Scherrer Institute, Villigen, Switzerland), operating in top-up mode at 2.4-GeV electron energy and a current of 400 mA. XAS data were collected at the Mo K-edge using a Si (111) monochromator in transmission mode between 19,800 and 21,150 eV with a step size of 0.25 eV. The calibration of the XAS data was based on the Mo foil at 20,000 eV. The sample was pressed into a pellet with an optimized amount of sample mixed with cellulose and sealed in air-tight aluminized plastic bags in the glovebox. The processing of the XAS data was performed with ProQEXAFS and Athena software[67,68]. The EXAFS fitting was conducted with the Artemis software[68]. The fitted variables include the CN, interatomic distance $R$, bond length disorder factors (Debye–Waller factors, DW), and energy shift. The amplitude reduction factor $S_0^2 = 0.96$ was obtained from fitting the corresponding Mo foil. Data fitting was carried out in the range of 1.0–3.0 Å and with a window $\Delta R$ of 0.5; the Fourier transform was carried out for $k = 3.0-15.0$ Å$^{-1}$.

TEM measurements were performed on an FEI Talos F200X transmission electron microscope operated at 200 kV. The STEM measurements were carried out in the same instrument with a resolution of 0.16 nm and a high-angle annular dark-field (HAADF) detector. The energy-dispersive X-ray spectroscopy (EDX) was obtained with a Super-X EDS system (windowless, shutter protected). SEM measurements were conducted on an FEI Magellan 400 FEG microscope (0.05–30 kV) with an EDAX Octane Elect Super EDS System. Before the measurement, the sample was sputter-coated with a ca. 5 nm layer of PtPd.

H$_2$ TPR, NH$_3$ TPD, CO TPD, H$_2$ TPD, and CO chemisorption were performed on an AutoChem (Micromeritics) instrument with a TCD. Ca. 100 mg of the specimen was loaded in a U-shape quartz reactor. The H$_2$ TPR was performed under 5% H$_2$/Ar from room temperature to 500 °C with a heating rate of 10 °C min$^{-1}$. In a typical NH$_3$ TPD experiment, the sample was pretreated at 300, 500, or 700 °C under pure H$_2$ for 2 h, saturated in 5% NH$_3$/He flow for 30 min at 50 °C, and purged with He for another 30 min. After that, the sample was heated to 1000 °C at 10 °C min$^{-1}$ under He flow and the desorbed NH$_3$ was monitored with the TCD detector. A similar experiment without NH$_3$ introduction was conducted to obtain the background. CO TPD and H$_2$ TPD were performed following similar procedures from −50 °C. For CO chemisorption, the sample was first pretreated at 300, 500, or 700 °C under pure H$_2$ for 2 h, and the CO adsorption isotherm were acquired at 0 °C.

The TGA-TPR study was performed in a Mettler Toledo TGA-DSC 3+ under 10% H$_2$/N$_2$ from room temperature to 500 °C at the heating rate of 5 °C min$^{-1}$. The temperature-programmed oxidization (TPO) study in a TGA was performed under air from room temperature to 800 °C using a heating rate of 5 °C min$^{-1}$. Raman spectroscopy was performed in a DXR 2 Raman spectrometer (Thermo Fisher) using a 532 nm excitation laser. During the measurement, the sample was loaded in an in situ cell (Linkam CCR1000) with flowing N$_2$ to protect the sample from damage by the laser. For the operando Raman study, the sample was first pretreated in pure H$_2$ at 500 °C for 2 h, followed by flow H$_2$/CO$_2$/N$_2$ (3/1/1) under 1 bar at 430 °C for 2.5 h.

The $^{95}$Mo solid-state NMR spectra were obtained at a principal magnetic field of 20.0 T (i.e., a Larmor frequency of 55.1 MHz) using a 4 mm diameter rotor, spinning at 14.3 kHz, and with a temperature set at 5 °C. The radio-frequency field used was 25 kHz, leading to an optimum 90° pulse of 5.0 µs and a CPMG sequence summing 30 echos was able to significantly increase the signal-to-noise ratio. Under those conditions, a Hahn Echo sequence did not provide a sufficiently large irradiation bandwidth to fully cover the extent of the spectra and we applied a VOCS procedure[69,70], recording 11 spectra separated by a 50 kHz offset. In total, 16k scans were acquired with a recycle delay of 0.5 s for each sub-spectrum. The spectra are referenced to a 2 M solution of Na$_2$MoO$_4$. Further details are provided in the Supplementary Information file.

**Catalytic testing.** The CO$_2$ hydrogenation reactions were conducted in a high-pressure tubular reactor (304.8 mm of length, 9.1 mm of internal diameter, Has-telloy X, Microactivity Effi, PID Eng&Tech) as reported in our previous study[71].

For a typical reaction, the catalyst (50 mg) was loaded in the glovebox and transferred without exposure to air. The H$_2$ treated catalysts Mo$_2$CT$_{x\text{-}300}$ and 2D-Mo$_2$C were prepared in situ prior to the catalytic tests under an H$_2$ flow (50 mL min$^{-1}$, 10 °C min$^{-1}$, 2 h). Before the reaction, the catalyst was protected under N$_2$ to the designated temperature (10 °C min$^{-1}$, 130–430 °C) and charged to the designated pressure (1–25 bar) with N$_2$. The gas feed was then switched to the reaction gas mixture of H$_2$, CO$_2$, and N$_2$ with a specific H$_2$/CO$_2$ ratio (1/2, 1/1, or 3/1) and 20 vol% N$_2$ as a balance. The products were analyzed online by double-channel gas chromatography (PerkinElmer Clarus 580) with the transfer line heated to 150 °C. H$_2$, N$_2$, and CO$_2$ were analyzed in Channel A equipped with a RESTEK ShinCarbon ST Micropacked Column and a TCD. CO, CH$_4$, methanol, DME, and C$_2$–C$_5$ hydrocarbons were analyzed in Channel B with an Agilent HP-PLOT Q Column, a methanizer, and a flame ionization detector. Different contact times (space velocities) were probed by changing the gas flow rate from 100 to 15 NmL min$^{-1}$. The product formation rate, CO$_2$ conversion, and selectivity to the given product were calculated with the following equations:

$$F_{x,\text{out}} \left[\text{mol h}^{-1}\right] = \frac{C_{x,\text{out}} \times F_{N_2,\text{in}}}{C_{N_2,\text{out}}} \tag{1}$$

$$r_x \left[\text{g h}^{-1} \text{g}_{\text{cat}}^{-1}\right] = \frac{F_{x,\text{out}}}{m_{\text{cat}}} \times \text{MW}_x \tag{2}$$

$$X_{CO_2} = \frac{\sum_{i=1}^{n} F_{x,\text{out}}}{F_{CO_2,\text{in}}} \tag{3}$$

$$S_x = \frac{F_{x,\text{out}}}{\sum_{i=1}^{n} F_{x,\text{out}}} \tag{4}$$

where $F_{x,\text{out}}$ is the outlet flow rate of species $x$ [mol h$^{-1}$]; $C_{x,\text{out}}$ is the outlet gas fraction of species $x$; $F_{x,\text{in}}$ is the inlet flow rate of species $x$ [mol h$^{-1}$]; $r_x$ is the formation rate of species $x$ [g h$^{-1}$ g$_{\text{cat}}^{-1}$]; $m_{\text{cat}}$ is the mass of catalyst used in the reaction [g]; MW$_x$ is the molecular weight of species $x$ [g mol$^{-1}$]; $X_{CO2}$ is the conversion of CO$_2$; $S_x$ is the selectivity of species $x$. Intrinsic formation rates were extrapolated using a second-order polynomial fit to the experimental data. Intrinsic selectivities were calculated from the intrinsic formation rates.

The WGS reaction was performed in a fixed-bed quartz reactor (400 mm of length and 12.6 mm of internal diameter) at atmospheric pressure. Mo$_2$CT$_x$ (30 mg) was supported on a plug of glass wool and the 2D-Mo$_2$C catalyst was prepared in situ prior to the catalytic test. The WGS reaction was performed under a stream of CO, H$_2$O, and N$_2$ (total flow rate 55 mL min$^{-1}$, CO/H$_2$O/N$_2$ = 1/1/9) at 500 °C for 10 h. The steam was generated from water evaporation with an evaporation mixer (Bronkhorst) and the water flow rate was controlled by a liquid flow meter (Bronkhorst, µ-Flow series). The off-gas after condensation of the unreacted steam was analyzed by double-channel gas chromatography (PerkinElmer Clarus 580) with thermal conductivity and flame ionization detectors.

The CO$_2$ dissociation experiments were performed in a tubular reactor. At 430 °C, the catalysts were exposed to 2% CO$_2$/N$_2$ (100 mL min$^{-1}$) for 10 min, purged with N$_2$ (100 mL min$^{-1}$) for 10 min, and reactivated with 10% H$_2$/N$_2$ (100 mL min$^{-1}$). The gases were measured online using a gas analyzer (ABB, EL3020) with a frequency of 1 Hz. A similar experiment was performed in a TGA (Mettler Toledo). Here, ca. 50 mg of Mo$_2$CT$_x$ was loaded in a sapphire crucible and treated in 10% H$_2$/N$_2$ (100 mL min$^{-1}$) at 500 °C for 2 h (10 °C min$^{-1}$). The sample was then cooled to 430 °C in N$_2$ and treated with 10% CO$_2$/N$_2$ for 6 h. After purging with N$_2$ for 10 min, the sample was treated with 10% H$_2$/N$_2$ (100 mL min$^{-1}$) for another 6 h. Three cycles were performed between 10% CO$_2$/N$_2$ and 10% H$_2$/N$_2$.

## Data availability

The data supporting the findings of this study are available from the corresponding authors upon reasonable request.

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

## Acknowledgements

This project has received funding from the European Union's Horizon 2020 research and innovation program (grant agreement No 800419) as well as from the ETH Zürich (grants ETH-40 17-2, ETH-44 16-2, and ETH-40 19-2) and from the Stavros Niarchos Foundation. The authors thank ScopeM (ETH Zürich) for the use of their electron microscopy facilities and the Laboratory of Surface Science and Technology (LSST, ETH Zürich) for the use of their XPS facilities. We acknowledge PSI SuperXAS for beamtime and thank Dr. Olga Safonova for assistance. We thank Dr. Agnieszka Kierzkowska for conducting the SEM imaging and Dr. Felix Donat for assistance with the in situ XRD measurement.

## Author contributions

A.F. conceived the research project. H.Z. planned the research. Z.C., E.K. and H.Z. prepared materials. H.Z. characterized and tested the catalysts and analyzed the data. A.T. collected XPS data. H.Z., Z.C. and P.M.A. performed XAS experiments. P.M.A. supervised XAS experiments. P.F. performed and analysed the solid-state NMR experiments. A.F. and C.R.M. coordinated the research. The data were discussed among all coauthors. H.Z. and A.F. wrote the paper with contributions from all authors.

## Competing interests

The authors declare no competing interests.
