## [Peer Review File · Nature Communications]

Two-Dimensional Molybdenum Carbide 2D-Mo₂C as a Superior Catalyst for CO₂ HydrogenationREVIEWER COMMENTS

Reviewer #1 (Remarks to the Author):

This is a very interesting study presenting a new type of molybdenum carbide catalysts for CO₂ hydrogenation. I like the general scope of the study, but some key points need better clarification:

- 1) On page 5, can the structures of Mo₂CT_x and 2D-Mo₂C be switched depending on the concentration of Tx in the chemical environment? Is the transformation seen in Figure 1a reversible?
- 2) On page 12, the new catalyst has a high long-term stability, and its conversion is comparable to that of Cu/ZnO/Al₂O₃, but there is also the issue of selectivity towards methanol production. I see a wide range of products on the 2D-Mo₂C catalyst. Can this be tuned depending on the nature and concentration of Tx?
- 3) On page 13, are the authors proposing a carbide or an oxycarbide catalyst? What is the best terminology to describe the system?
- 4) On page 19, at this point it will be useful to present a discussion of the catalytic process at a microscopic level and how it compares with regular carbide- and oxide-based catalysts. What are the key sites in the 2D-Mo₂C system and how they affect its performance?

Reviewer #2 (Remarks to the Author):

The manuscript by Zhou et al. describes an elegant study in which a layered Mo₂C catalyst is synthesized from a MAXene phase and investigated for CO₂ hydrogenation activity.

The manuscript is very well written and easy to read. The amount and quality of data provided, both kinetic and material characterization data, is extraordinary. Still the authors manage to combine and connect the different experiments clearly without losing the reviewer/reader, despite 62 figures in the supplementary. This warrants a heartfelt congratulation.

The manuscript is already on a very high standard but could even further be improved with some minor changes:

- Figure 1 e is very difficult to follow and maybe the authors should consider a top view of the in situ XRD data.
- It would be interesting to hear the opinion of the authors if the observed lesser activation of the MAXene phase in diluted hydrogen could be compensated at higher pressures, i.e. at equivalent H₂ partial pressures.
- The conclusions of the in situ Raman study could be enhanced by providing a spectra at 25C after the heat treatment to confirm that the E_{2g} peak actually disappears and the observation is not based on the increased temperature.
- While the authors provide lattice parameters from XRD it would be interesting to understand exactly what the layer thickness and the inter layer distance is in the 2D Mo₂C.
- The difference in activity of 2D Mo₂C to beta Mo₂C is mostly related to the CO chemisorption results. This data is nearly equated to the surface area. However the authors also comment on the preferred surface termination of 2D Mo₂C. A non reactive surface area measurement such as BET could add to the picture and show if the activity per surface area is actually higher in 2D Mo₂C due to the preferred surface termination. Otherwise beta Mo₂C could 'simply' be synthesized as nanoparticles to match the 2D Mo₂C.
- The authors should comment on the apparent reduction of product formation rates (for example CO in Fig S28) although no deactivation through oxidation is reported.

- The thermodynamic calculations should be explained a bit better. What do they represent? Is it a minimization of the Gibbs free energy of the system of equilibrium conversions of different mixtures of H₂ and CO₂ for the individual reactions? If the latter the independence of CO yield with H₂:CO₂ ratio is against previously reported calculations which show an increased CO₂ conversion on the RWGS. In Fig 2b also just a single thermodynamic equilibrium conversion is provided. It is assumed that this is for the RWGS and it should change with feed ratio.

-The authors observe that 2D Mo₂C becomes a methanation catalyst under RWGS conditions at elevated pressures. While it is correct, that this increases flexibility of the catalyst, there are clearly more effective methanation catalysts available. The question why this methanation activity is not observed for beta Mo₂C is unfortunately not discussed. When thinking of using RWGS as an industrial pathway to activate CO₂ industrially, this would likely have to be conducted at elevated pressures to increase efficiency and due to possible down stream process requirements.

The reviewer would again like to thank the authors for their excellent contribution and is looking forward to their publication.

Reviewer #3 (Remarks to the Author):

Zhou and coworkers show here, in a fully documented study, that Mo₂C Mxene material is potentially a remarkable catalyst for the CO₂ conversion into other useful chemicals (such as CO, CH₄, methanol, dme, or alkanes), particularly true when clean of surface terminations, quite stable over time and redox cycles, capable of reaching thermodynamic limits, and with tunable selectivity, which surpasses the activity of other Molybdenum carbide materials, and Cu-based commercial catalyst.

This is definitely a breakthrough study, which the MXene field was missing; this is, the use of MXenes in heterogeneous catalysis on CO₂ conversion, and will be, undoubtedly, a path-maker. I would gladly recommend publication, although there is a number of points that authors should regard prior to that.

1. Authors point out that orthorhombic b-Mo₂C has been reported for CO₂ hydrogenation, particularly in the RWGS. Quite relevant in this context, hexagonal alfa-Mo₂C has been pointed out as an excellent RWGS catalyst with almost full selectivity towards CO and also reaching the thermodynamic limit (10.1021/acscatal.7b00735). This study also reveals that the carbide does not form oxycarbide under H₂ environment, as here also shown.

2. When discussing Ref. 31, authors point out that unreduced MXene is a better than reduced for the WGS. However, their finding is the opposite (reduced MXene is better for the RWGS). Can authors elaborate this comparison, and reconcile the apparent opposite effects of reducing surface termination on the catalyzed reactions?

3. Authors stress her the gram-scale scalable synthesis protocol; in this regard it is relevant to point out the gram synthesis of Ti₃C₂T_x (10.1002/adem.201901241).

4. Ref. 27 already revealed that having a partial surface terminations, this is, some regions on unterminated MXenes, may be quite beneficial for the catalytic activity, there fro the DRM reaction. This is quite related to this study, and so, the parallelism should be highlighted.

5. There is a previous report (10.1002/adma.201805472) in which Ti₃C₂T_x is also reduced by annealing and H₂ treatment to remove F and O terminations. There, however, F is firstly eliminated by annealing at 650°C, while O (and OH) removed at 700°C under H₂ atmosphere, contrary to here where OH is alleged to be removed first, and F being the most resisting termination. Citation and comparision and discussion is a must.

6. After H₂ treatment; how sure are authors on having pristine surface, with no H-termination?

7. One important aspect, not really highlighted in the article, are the high degrees of CO₂ conversion, in the dozens of mg per h and gram. This contrasts with previous studies on the CO₂ reduction by electrocatalytic (10.1016/j.isci.2020.101181) or photocatalytic means, combined with semiconductors (10.1002/cssc.201800083), where conversions of the order of micromoles are achieved. Thus, the use of MXenes as heterogeneous catalysts for a large CO₂ conversion has to be highlighted.

8. When comparing the TPD of 2D-Mo₂C and b-Mo₂C, a difference of only 12°C is little. As later discussed, both materials are similar, the major difference being the larger surface area. This has been signaled, e.g. in the CO₂ affinity, see 10.1039/C9CP04833B. A comment on this should be included.

9. It seems as a key aspect in CO₂ conversion and hydrogenation is that CO₂ chemically adsorbs, gets activated (Ref. 10 and 10.1039/C8CP02746C), and easily broken (Ref. 19). It also helps that Mo₂C systems tend to easily dissociate H₂ (10.1016/j.susc.2016.10.001).

10. Minor style point; use Oxford comma, use (0001) surface notation; put in italics foreign words (i.e., e.g., ca., versus, in situ). Do not put T in Tx in italics. Use better "whether" instead of "if" in a question.

Response to comments of Reviewer #1

This is a very interesting study presenting a new type of molybdenum carbide catalysts for CO₂ hydrogenation. I like the general scope of the study, but some key points need better clarification:

We thank Reviewer 1 for the generally positive evaluation of our work and for valuable comments. A point-by-point response to the critical comments of Reviewer 1 is provided below.

Q1. On page 5, can the structures of Mo₂CT_x and 2D-Mo₂C be switched depending on the concentration of T_x in the chemical environment? Is the transformation seen in Figure 1a reversible?

A1. The change of oxidation state of Mo (controlled by the T_x coverage, where T_x = O, OH, F) is indeed reversible, however the interlayer distance of the initial Mo₂CT_x (containing also intercalated water) is not recovered by the oxidation of 2D-Mo₂C (note that T_x = O, OH in the oxidized 2D-Mo₂C material). In particular, when we expose 2D-Mo₂C to conditions of the WGS reaction (CO+H₂O), the initial activity decreased significantly in the first 4 hours on stream (Supplementary Fig. 31). Mo K-edge XANES data reveal that the Mo state of this material (i.e. 2D-Mo₂C after the WGS reaction) is close to that of Mo₂C T_x (Supplementary Fig. 32). However, the lower activity of the materials compared to Mo₂CT_x is explained by the increased mass transport limitations due to the reduced interlayer distance.

Q2. On page 12, the new catalyst has a high long-term stability, and its conversion is comparable to that of Cu/ZnO/Al₂O₃, but there is also the issue of selectivity towards methanol production. I see a wide range of products on the 2D-Mo₂C catalyst. Can this be tuned depending on the nature and concentration of T_x?

A2. The selectivity of the products can be tuned by the coverage of the T_x groups and is influenced strongly by the reaction conditions. The long-term stability test, mentioned by Reviewer 1 above, was performed at 430 °C and 1 bar, and the formation of methanol is not favoured under these conditions. A high methanol selectivity was found at 130 °C (25 bar), i.e. 41% for Mo₂CT_x and 62% for 2D-Mo₂C (Supplementary Fig. 39). At a more common condition for methanol synthesis (230 °C, 25 bar) selectivities to methanol displayed by Mo₂CT_x and 2D-Mo₂C are 11% and 6%, respectively (Supplementary Fig. 39).

Q3. On page 13, are the authors proposing a carbide or an oxycarbide catalyst? What is the best terminology to describe the system?

A3. Pristine 2D-Mo₂C is best described as a carbide material. But when exposed to reactants (CO₂+H₂), it evolves into a 2D-Mo₂CO_x material (with the surface coverage of O_x groups dependent on the specific reaction conditions). 2D-Mo₂CO_x is therefore an oxycarbide catalyst. We verified in the text that the working state of 2D-Mo₂C is referred to as an oxycarbide, and the pristine state (i.e. prior to the exposure to the reactants) is referred to as a carbide state.

Q4. On page 19, at this point it will be useful to present a discussion of the catalytic process at a microscopic level and how it compares with regular carbide- and oxide-based catalysts. What are the key sites in the 2D-Mo₂C system and how they affect its performance?

A4. We agree with Reviewer 1 that a microscopic level discussion would be useful for understanding the reaction pathways on 2D-Mo₂CO_x, contrasting those to that on the 3D-Mo₂C catalyst (such as β-Mo₂C), and understanding the influence of the surface coverage effect on the reaction rates and selectivity. Our collaborators are performing DFT calculations in this direction, but this work merits a separate publication that will be reported in due course.

Response to comments of Reviewer #2

The manuscript by Zhou et al. describes an elegant study in which a layered Mo₂C catalyst is synthesized from a MAXene phase and investigated for CO₂ hydrogenation activity.

The manuscript is very well written and easy to read. The amount and quality of data provided, both kinetic and material characterization data, is extraordinary. Still the authors manage to combine and connect the different experiments clearly without losing the reviewer/reader, despite 62 figures in the supplementary. This warrants a heartfelt congratulation.

The manuscript is already on a very high standard but could even further be improved with some minor changes:.

We thank Reviewer 2 for recognizing a “very high standard” of our manuscript and referring to the results as “extraordinary”. We answer, point-by-point, the critical questions of Reviewer 2 below.

Q5. Figure 1 e is very difficult to follow and maybe the authors should consider a top view of the in situ XRD data.

A5. We have added a 3D-view of the in situ XRD pattern in the Supplementary Fig. 9 (shown below) since the top view plot did not improve the clarity as expected. The respective note about Supplementary Fig. 9 has also been added to the caption of Figure 1.

Supplementary Fig. 9. Reduction of Mo_2CT_x followed by in situ XRD.

Q6. It would be interesting to hear the opinion of the authors if the observed lesser activation of the MAXene phase in diluted hydrogen could be compensated at higher pressures, i.e. at equivalent H_2 partial pressures.

A6. We think it is reasonable to expect that higher pressures will compensate for the use of diluted hydrogen, providing effectively the same partial pressure of H_2 (that is, a total of 10 bar of 10% H_2/N_2), although an optimization of the contact time may be required.

Q7. The conclusions of the in situ Raman study could be enhanced by providing a spectra at 25C after the heat treatment to confirm that the E_{2g} peak actually disappears and the observation is not based on the increased temperature.

A7. We did not record the Raman spectrum at room temperature after the in situ reduction experiment because cooling down could have caused condensation of water, physisorbed on the interior surfaces of the in situ cell, back on the reduced highly oxophilic 2D- Mo_2C , and this water could possibly oxidize the surface. We have compared the results of our in situ Raman study of Mo_2CT_x under hydrogen with literature results under argon reported in Chem. Mater. 2017, 29, 6472, where the E_{2g} peak is preserved during the in situ Raman study under Ar. Therefore, the disappearance of the E_{2g} peak under hydrogen due to the temperature effect is unlikely. Together with other characterization methods (TPR, NMR, XRD, XAS, and XPS), we believe it is reasonable to conclude that the disappearance of the E_{2g} peak is due to the removal of T_x groups. We added a comment on the comparison of in situ Raman results under Ar and H_2 in the manuscript and cited the reference below.

Added ref: Kim, H., Anasori, B., Gogotsi, Y. & Alshareef, H. N. Thermoelectric Properties of Two-Dimensional Molybdenum-Based MXenes. Chem. Mater. 29, 6472–6479 (2017).

Q8. While the authors provide lattice parameters from XRD it would be interesting to understand exactly what the layer thickness and the inter layer distance is in the 2D Mo_2C .

A8. So far, we have not been able to properly refine the atomic positions in the structure of 2D-Mo₂C by the Rietveld refinement of the XRD data. (This analysis is likely hindered by the high anisotropy of the peaks as well as by the static disorder). However, using the Le Bail fitting we can assign the space group as the same as the parent MAX phase, and determine lattice parameters (but not the atomic positions). Although we could not refine the atomic positions, one can assume that Mo and C remain in the same position as in the parent MAX phase. Assuming such structure, an approximate calculation of the spacing between the layers is possible, which can be estimated as ca. 5 Å. We have added a comment in the revised Manuscript text that the interlayer distance can be roughly estimated as ca. 5 Å.

Q9. The difference in activity of 2D Mo₂C to beta Mo₂C is mostly related to the CO chemisorption results. This data is nearly equated to the surface area. However the authors also comment on the preferred surface termination of 2D Mo₂C. A non reactive surface area measurement such as BET could add to the picture and show if the activity per surface area is actually higher in 2D Mo₂C due to the preferred surface termination. Otherwise beta Mo₂C could 'simply' be synthesized as nanoparticles to match the 2D Mo₂C.

A9. We agree with the argument of Reviewer 2. However, the non-reactive surface area of 2D-Mo₂C and Mo₂C_{T_x} is difficult to measure due to the 2D structure of these materials and the low interlayer distance (note that for this reason, literature reports on MXene phases usually do not contain information on the BET-determined surface area). Therefore, we rely on CO chemisorption as a way to quantify the number of exposed Mo atoms and compare the catalytic activity.

Q10. The authors should comment on the apparent reduction of product formation rates (for example CO in Fig S28) although no deactivation through oxidation is reported.

A10. Please note that the x-axis in figures such as Fig. S28 is contact time instead of time on stream (TOS) as implied by Reviewer 2 in Q10. The contact time is defined as the residence time of the reaction stream in the catalyst bed, which is inversely proportional to the space velocity. Therefore, the increase of contact time means the decrease of gas flow rate of CO₂ and H₂, and thus the formation rate of CO is decreased with a decreasing feeding rate of CO₂.

Q11. The thermodynamic calculations should be explained a bit better. What do they represent? Is it a minimization of the Gibbs free energy of the system of equilibrium conversions of different mixtures of H₂ and CO₂ for the individual reactions? If the latter the independence of CO yield with H₂:CO₂ ratio is against previously reported calculations which show an increased CO₂ conversion on the RWGS. In Fig 2b also just a single thermodynamic equilibrium conversion is provided. It is assumed that this is for the RWGS and it should change with feed ratio.

A11. Yes, the thermodynamic calculation is a minimization of the Gibbs free energy of a system of pre-specified (potential) products and reactants. This information has been added in the figure captions in the manuscript and the Supplementary Information. Reactions are not required to be specified during the thermodynamic calculation. The CO yield depends on the H₂/CO₂ ratio. In Fig. 2b, the thermodynamic equilibrium conversion is plotted for a H₂/CO₂ feed ratio of 3:1. We added this information for clarification.

Q12. The authors observe that 2D Mo₂C becomes a methanation catalyst under RWGS conditions at elevated pressures. While it is correct, that this increases flexibility of the catalyst, there are clearly more effective methanation catalysts available. The question why this methanation activity is not observed for beta Mo₂C is unfortunately not discussed. When thinking of using RWGS as an industrial pathway to activate CO₂ industrially, this would likely have to be conducted at elevated pressures to increase efficiency and due to possible down stream process requirements.

A12. We agree with Reviewer 2 that this would be an interesting question to understand. However, it will require microkinetic modelling, which is not available at this stage. We hope to clarify this question in our subsequent studies.

The reviewer would again like to thank the authors for their excellent contribution and is looking forward to their publication.

We thank Reviewer 2 for comments and for constructive suggestions on our manuscript.

Response to comments of Reviewer #3

Zhou and coworkers show here, in a fully documented study, that Mo₂C MXene material is potentially a remarkable catalyst for the CO₂ conversion into other useful chemicals (such as CO, CH₄, methanol, dme, or alkanes), particularly true when clean of surface terminations, quite stable over time and redox cycles, capable of reaching thermodynamic limits, and with tunable selectivity, which surpasses the activity of other Molybdenum carbide materials, and Cu-based commercial catalyst.

This is definitely a breakthrough study, which the MXene field was missing; this is, the use of MXenes in heterogeneous catalysis on CO₂ conversion, and will be, undoubtedly, a path-maker. I would gladly recommend publication, although there is a number of points that authors should regard prior to that.

We thank Reviewer 3 for the positive evaluation of our work, recognizing it as "a breakthrough study" and "a path-maker". A point-by-point response to critical comments of Reviewer 3 is provided below.

Q13. Authors point out that orthorhombic b-Mo₂C has been reported for CO₂ hydrogenation, particularly in the RWGS. Quite relevant in this context, hexagonal alpha-Mo₂C has been pointed out as an excellent RWGS catalyst with almost full selectivity towards CO and also reaching the thermodynamic limit (10.1021/acscatal.7b00735). This study also reveals that the carbide does not form oxycarbide under H₂ environment, as here also shown.

A13. We have included the relevant citation mentioned by Reviewer 3 in the revised Introduction:

In this context, Mo₂C has been reported as a promising catalyst for CO₂ hydrogenation, yielding a particularly high selectivity to CO via the reverse water-gas shift (RWGS) reaction^{9,10,18-20}.

Q14. When discussing Ref. 31, authors point out that unreduced MXene is a better than reduced for the WGS. However, their finding is the opposite (reduced MXene is better for the RWGS). Can authors elaborate this comparison, and reconcile the apparent opposite effects of reducing surface termination on the catalyzed reactions?

A14. In this study, we show that 2D-Mo₂C is a better RWGS catalyst than Mo₂C T_x (in various conditions). We also compared the activity of Mo₂C T_x and 2D-Mo₂C for WGS at 500 °C. Mo₂C T_x shows a stable activity for at least 10 hours of TOS. In contrast, while the initial activity of 2D-Mo₂C is similar to that of Mo₂C T_x, it decreased significantly in the first 4 hours (Supplementary Fig. 31). XANES data reveals that 2D-Mo₂C is oxidized under WGS conditions, reaching after 10-hour of TOS an oxidation state (and therefore a T_x surface coverage) close to that of Mo₂C T_x (Supplementary Fig. 32). Owing to this similar T_x surface coverage, the lower activity of 2D-Mo₂C after 4 h of TOS is consistent with mass transport limitations due to the reduced interlayer distance in the H₂-pretreated Mo₂C T_x materials. The decrease of the activity of 2D-Mo₂C in the first 4 h of TOS is therefore linked with the surface oxidation (at 500 °C, likely, by steam) of 2D-Mo₂C, including the interlayer Mo sites. Note that such significant oxidation of 2D-Mo₂C under WGS conditions contrasts substantially the only minor oxidation of 2D-Mo₂C in RWGS conditions.

Q15. Authors stress her the gram-scale scalable synthesis protocol; in this regard it is relevant to point out the gram synthesis of Ti₃C₂T_x (10.1002/adem.201901241).

A15. Please note that in no way we dispute the gram-scale synthesis of MXenes in general as, in addition to the reference mentioned by Reviewer 3, many other literature reports have demonstrated the gram-scale synthesis of MXenes. Rather, we stress that owing to the available gram-scale synthesis of MXenes, it is possible to synthesize 2D-Mo₂C (free of termination groups) on gram scale as well. We discuss in the main text that "typical approaches to yield 2D-Mo₂C exploited so far chemical vapor deposition onto flat substrates and lead to ultra-thin orthorhombic Mo₂C," and we contrast the present method to synthesize 2D-Mo₂C to these previous reports that only yield limited amounts of 2D-Mo₂C. To avoid confusion, we have modified the introduction text as follows:

Here, enabled by the scalable synthesis of MXenes (ref), we report a gram-scale synthesis of a phase-pure multilayered hexagonal 2D-Mo₂C material with only Mo-terminated basal planes.

Added ref: Shuck, C. E. & Gogotsi, Y. Taking MXenes from the lab to commercial products. Chem. Eng. J. 401, 125786 (2020).

Q16. Ref. 27 already revealed that having a partial surface terminations, this is, some regions on unterminated MXenes, may be quite beneficial for the catalytic activity, there fro the DRM reaction. This is quite related to this study, and so, the parallelism should be highlighted.

A16. The active state in the DRM reaction, when using silica-supported 2D-Mo₂CO_x, features the oxidation state of Mo as +4, which corresponds to a partial surface coverage with oxygen atoms (the respective DFT model used 0.67 ML coverage of oxygen). This is in contrast to the most active state of the 2D-Mo₂C material for the CO₂ hydrogenation in this study, which oxidizes only moderately under RWGS reaction conditions, and therefore has a significantly lower than 0.67 ML surface coverage of oxygen. We added the following text:

This result is also different from that found in a recent DRM study, where an oxycarbide 2D-Mo₂CO_x with an average Mo oxidation state of +4 (modelled by the 2/3 of a monolayer oxygen surface coverage) is the active phase.

Q17. There is a previous report (10.1002/adma.201805472) in which Ti₃C₂T_x is also reduced by annealing and H₂ treatment to remove F and O terminations. There, however, F is firstly eliminated by annealing at 650°C, while O (and OH) removed at 700°C under H₂ atmosphere, contrary to here where OH is alleged to be removed first, and F being the most resisting termination. Citation and comparison and discussion is a must.

A17. We have revised the text as follows:

“The reduction steps at 280 °C likely are associated with the reductive defunctionalization of the surface hydroxyl groups and possibly, fluorine groups, while the peak at 500 °C is related to the removal of surface fluoro and oxo groups^{34,46,47}.” We have also added the reference requested by Reviewer 3. Note, however, that the oxophilicity of Ti and Mo is different (earlier transition metals are more oxophilic), and therefore defunctionalization temperatures (and the order of defunctionalization of oxo, oxy and fluoro groups) may vary for various MXenes.

Q18. After H₂ treatment; how sure are authors on having pristine surface, with no H-termination?

A18. We thank Reviewer 3 for this question. The surface H-termination of Mo can be stable at low temperatures. According to the H₂ temperature-programmed desorption (TPD) presented in Supplementary Fig. 28, the H₂ is desorbed only above 450 °C. However, 2D-Mo₂C is reduced under H₂ at 500 °C and cooled down under N₂, therefore the H-termination is not expected to be stable. Therefore, we believe there is no H-termination in the 2D-Mo₂C.

Q19. One important aspect, not really highlighted in the article, are the high degrees of CO₂ conversion, in the dozens of mg per h and gram. this contrasts with previous studies on the CO₂ reduction by electrocatalytic (10.1016/j.isci.2020.101181) or photocatalytic means, combined with semiconductors (10.1002/cssc.201800083), where conversions of the order of micromoles are achieved. Thus, the use of MXenes as heterogeneous catalysts for a large CO₂ conversion has to be highlighted.

A19. We added the comparison of the thermocatalytic CO₂ conversion in this study with the electrocatalytic or photocatalytic CO₂ conversion in the references to highlight the high conversion rate of CO₂ in this study as follows:

The product formation rates are much higher than those from previous studies on the CO₂ reduction by MXene-based electro¹ or photocatalysts².

References

1. Handoko, A. D. et al. Two-Dimensional Titanium and Molybdenum Carbide MXenes as Electrocatalysts for CO₂ Reduction. *iScience* 23, 101181 (2020).

2. Ye, M., Wang, X., Liu, E., Ye, J. & Wang, D. Boosting the Photocatalytic Activity of P25 for Carbon Dioxide Reduction by using a Surface-Alkalinized Titanium Carbide MXene as Cocatalyst. *ChemSusChem* 11, 1606–1611 (2018).

Q20. When comparing the TPD of 2D-Mo₂C and β-Mo₂C, a difference of only 12°C is little. As later discussed, both materials are similar, the major difference being the larger surface area. This has been signaled, e.g. in the CO₂ affinity, see 10.1039/C9CP04833B. A comment on this should be included.

A20. We agree that the major difference between the 2D-Mo₂C and β-Mo₂C comes from the different surface area, more specifically, the Mo surface area. This is also indicated from the CO chemisorption analysis (Table 1). We have modified the manuscript and included the suggested reference.

2D-Mo₂C (prepared in situ prior to CO chemisorption analysis) shows a significantly increased CO capacity of 41.1 μmol g⁻¹, which exceeds the CO capacity of β-Mo₂C (also reduced in H₂ at 500 °C in

situ) by ca. a factor of eight (Table 1). This result is explained by the larger specific surface area of the exposed Mo-terminated facets in 2D-Mo₂C, which is also consistent with the similar CO₂ adsorption energy of MXenes and 3D transitional metal carbides (Phys. Chem. Chem. Phys. 2019, 21, 23136).

Added ref: Morales-García, Á., Mayans-Llorach, M., Viñes, F. & Illas, F. Thickness biased capture of CO₂ on carbide MXenes. Phys. Chem. Chem. Phys. 21, 23136–23142 (2019).

Q21. It seems as a key aspect in CO₂ conversion and hydrogenation is that CO₂ chemically adsorbs, gets activated (Ref. 10 and 10.1039/C8CP02746C), and easily broken (Ref. 19). It also helps that Mo₂C systems tend to easily dissociate H₂ (10.1016/j.susc.2016.10.001).

A21. We agree with the reviewer that the key aspect is CO₂ adsorption, dissociation, and H₂ dissociation. We have revised the introduction by adding the suggested references.

Q22. Minor style point; use Oxford comma, use (0001) surface notation; put in italics foreign words (i.e., e.g., ca., versus, in situ). Do not put T in T_x in italics. Use better "whether" instead of "if" in a question.

A22. We agree and applied those suggested elements of style with two exceptions. According to the style guide of *Nat. Commun.*, foreign words should not use italics. In addition, T in T_x should be used in italics to comply with IUPAC regulations and not suggest, by the use of T in regular font, tritium-terminated surface.

REVIEWERS' COMMENTS

Reviewer #1 (Remarks to the Author):

The authors have improved considerably the scientific level of the paper. In general, the response of the authors to my previous comments is satisfactory, but there is a point that still needs to be clarify. In the the main text, the authors need to add a sentence that the activity and selectivity of molybdenum carbide nanoparticles towards CO₂ hydrogenation can be affected by the C/Mo ratio in the system (Figueras et al, ACS Catal. 2021, 11, 9679; Hamdan et al, Frontiers of Chemistry, June 2020, vol 8, article 452).

I recommend acceptance after minor revisions.

Reviewer #2 (Remarks to the Author):

The authors have fully addressed all previously raised concerns and I would like to support publication.

Reviewer #3 (Remarks to the Author):

The authors duly treated all the raised points. I am glad to recommend publication of the manuscript in its present form.

Response to comments of Reviewer #1

The authors have improved considerably the scientific level of the paper. In general, the response of the authors to my previous comments is satisfactory, but there is a point that still needs to be clarify. In the the main text, the authors need to add a sentence that the activity and selectivity of molybdenum carbide nanoparticles towards CO₂ hydrogenation can be affected by the C/Mo ratio in the system (Figueras et al, ACS Catal. 2021, 11, 9679; Hamdan et al, Frontiers of Chemistry, June 2020, vol 8, article 452).

I recommend acceptance after minor revisions.

We thank Reviewer #1 for recommending acceptance after minor revision. We have added the following sentence with the suggested references.

The activity and selectivity of CO₂ hydrogenation catalysts based on molybdenum carbide can be affected by the C/Mo ratio in the catalyst^{19,20}.

Added references

19. Figueras, M. *et al.* Supported Molybdenum Carbide Nanoparticles as an Excellent Catalyst for CO₂ Hydrogenation. *ACS Catal.* **11**, 9679–9687 (2021).

20. Abou Hamdan, M. *et al.* Supported Molybdenum Carbide and Nitride Catalysts for Carbon Dioxide Hydrogenation. *Front. Chem.* **8**, 452 (2020).